# Data Parameters: A New Family of Parameters for Learning a Differentiable Curriculum

**Shreyas Saxena**
Apple
shreyas_saxena@apple.com

**Oncel Tuzel**
Apple
otuzel@apple.com

**Dennis DeCoste**
Apple
ddecoste@apple.com

## Abstract

Recent works have shown that learning from easier instances first can help deep neural networks (DNNs) generalize better. However, knowing which data to present during different stages of training is a challenging problem. In this work, we address this problem by introducing *data parameters*. More specifically, we equip each sample and class in a dataset with a learnable parameter (data parameters), which governs their importance in the learning process. During training, at each iteration, as we update the model parameters, we also update the data parameters. These updates are done by gradient descent and do not require hand-crafted rules or design. When applied to image classification task on CIFAR10, CIFAR100, WebVision and ImageNet datasets, and object detection task on KITTI dataset, learning a dynamic curriculum via data parameters leads to consistent gains, without any increase in model complexity or training time. When applied to a noisy dataset, the proposed method learns to learn from clean images and improves over the state-of-the-art methods by $14\%$. To the best of our knowledge, our work is the first curriculum learning method to show gains on large scale image classification and detection tasks. Code is available at: `https://github.com/apple/ml-data-parameters`.

## 1 Introduction

*Curriculum learning* [1, 7, 12, 17, 35] has garnered lot of attention in the field of machine learning. It draws inspiration from the learning principles underlying cognitive process of humans and animals, which starts by learning easier concepts and then gradually transitions to learning more complex concepts. Existing work has shown that with the help of this paradigm, DNNs can achieve better generalization [1, 2, 15].

The key to applying curriculum learning to different problems is to come up with a ranking function that assigns learning priorities to the training samples. A sample with a higher priority is supposed to be learned earlier than a sample with a lower priority. For the majority of early work in curriculum learning, the curriculum is provided by a pre-determined heuristic. For instance, for the task of classifying shapes [1], shapes which had less variation were assigned a higher priority. In [29], authors approached grammar induction, where short sentences were assigned higher priority. The main issues which limit the application of this approach are: (1) for many complex problems, it is not trivial to define what are the easy examples or subtasks, (2) in cases where humans can design a curriculum, it is assumed that the difficulty of learning a sample for humans correlates with the difficulty of learning the sample for a learning algorithm, and (3) even if one could define the curriculum, the pre-determined curriculum might not be appropriate at all learning stages of the dynamically learned model.

Learning a curriculum in an automatic manner is a hard task, since the ease or difficulty of an example is relative to the current state of the model. In order to overcome these issues, in this work, we

introduce a new family of parameters for DNNs termed *data parameters*. More specifically, each class and data point have their own *data parameter*, governing their importance in the learning process. During learning, at every iteration, as we update the standard model parameters, we also update the data parameters using stochastic gradient descent. Learning data parameters for class and instances leads to a dynamic and differentiable curriculum, without any need of human-intervention.

The main contributions of our work are:

1. We introduce a new class of parameters termed *data parameters* for every class and data point in the dataset. We show that data parameters can be learned using gradient descent, and doing so amounts to learning a dynamic and differentiable curriculum. In our formulation, data parameters are involved only during training, and hence do not affect model complexity at inference.

2. We show that for image classification and object detection tasks, learning a curriculum for CNNs improves over baseline by prioritizing classes and their instances. To the best of our knowledge, our paper is the first curriculum learning method to show gains on large scale image classification tasks (ImageNet [5]) and on an object detection task (KITTI [9]).

3. We show that in presence of noisy labels, the learnt curriculum prioritizes learning from clean labels. Doing so, our method outperforms the state-of-art by a significant margin.

4. We show that when presented with random labels, in comparison to a baseline DNN which memorizes the data, the learned curriculum resists memorizing corrupt data.

## 2   Learning a Dynamic Curriculum via Data Parameters

As suggested earlier, the main intuition behind our idea is simple: each class and data point in the training set has a parameter associated to it, which weighs the contribution of that class or data point in the gradient update of model parameters. In contrast to existing works which set these parameters with a heuristic, in our work these parameters are learnable and are learnt along with the model parameters. Unlike model parameters which are involved during training and inference, data parameters are only involved during training. Therefore, using data parameters during training does not affect the model complexity and run-time at inference. In the next section we formalize this intuition for class-level curriculum, followed by instance-level curriculum.

### 2.1   Learning curriculum over classes

We first describe learning a dynamic curriculum over classes where the contribution of each sample to the model learning is determined by its class. This curriculum favors learning from easier classes at the earlier stages of training. The curriculum over classes is dynamic and is controlled by the class-level data parameters, which are also updated via the training process. In what follows, we will refer to class-level data parameters as class-parameters.

Let $\left\{ \left( \mathbf{x}^i, y^i \right) \right\}_{i=1}^{N}$ denote the data, where $\mathbf{x}^i \in \mathbb{R}^d$ denotes a single data point and $y^i \in \{1, ..., k\}$ denotes its target label. Let $\sigma^{class} \in \mathbb{R}^k$ denote the class parameters for the classes in the dataset. We denote the neural network function mapping the input data $\mathbf{x}^i$ to logits $\mathbf{z}^i \in \mathbb{R}^k$ as $z^i = f_\theta(\mathbf{x}^i)$ where $\theta$ are the model parameters. During training, we pass the input sample $\mathbf{x}^i$ through the DNN, and compute its corresponding logits $\mathbf{z}^i$, but instead of computing the softmax directly on the logits, we scale the logits of the instance with parameter corresponding to the target class, $\sigma_{y^i}^{class}$. Note, scaling of logits with the parameter of target class can be interpreted as a temperature scaling of logits. The cross-entropy loss for a data point $\mathbf{x}^i$ can then be written as

$$L^i = -\log(p_{y^i}^i)$$
$$p_{y^i}^i = \frac{\exp(z_{y^i}^i / \sigma_{y^i}^{class})}{\sum_j \exp(z_j^i / \sigma_{y^i}^{class})} \tag{1}$$

where $p_{y^i}^i, z_{y^i}^i$ and $\sigma_{y^i}^{class}$ denote probability, logit and parameter of the target class $y^i$ for data point $\mathbf{x}_i$ respectively. If we set all class parameters to one, i.e. $\sigma_j^{class} = 1, j = 1 \ldots k$, we recover the gradient for the standard cross-entropy loss.

During training we solve

$$\min_{\theta, \sigma^{class}} \frac{1}{N} \sum_{i=1}^{N} L^i \tag{2}$$

where in addition to the model parameters, $\theta$, we also optimize the class-level parametres, $\sigma^{class}$.

The gradient of the loss with respect to logits is given by:

$$\frac{\partial L^i}{\partial z_j^i} = \frac{p_j^i - 1(j = y^i)}{\sigma_{y^i}^{class}} \tag{3}$$

where $1(j = y^i)$ means value 1 when $j = y^i$ and value 0 otherwise. The gradient of the loss with respect to the parameter of target class is given by:

$$\frac{\partial L^i}{\partial \sigma_{y^i}^{class}} = \frac{(1 - p_{y^i}^i)}{(\sigma_{y^i}^{class})^2} \Big( z_{y^i}^i - \sum_{j \neq y^i} q_j^i z_j^i \Big) \tag{4}$$

where $q_j^i = \frac{p_j^i}{1 - p_{y^i}^i}$ is the probability distribution over non-target classes (indexed by $j$, with $j \neq y^i$).

**Effect of class parameters on learning:** The class parameters are updated with the negative of the gradient given in equation (4), where the parameter corresponding to target class $\sigma_{y^i}^{class}$ will increase if the logit of the target class is less than the expected value of logits on non-target classes (i. e. $z_{y^i}^i < \sum_{j \neq y^i} q_j^i z_j^i$) and vice-versa. Therefore, during the course of learning, if data-points of a certain class are being misclassified, the gradient update on class parameters gradually increases the parameter associated with this class. Increasing the class parameter, flattens the curvature of the loss function for instances of that class, thereby decaying the gradients w.r.t. logits (see equation (3)). Decreasing the class parameter has an inverse effect, and accelerates the learning.

## 2.2 Learning curriculum over instances

In the previous section, we have detailed how we can learn a dynamic curriculum over classes of a dataset. A natural extension of this framework is to have a dynamic curriculum over the instances in the dataset. In this case, in equation (1), rather than having class parameters for each class $\sigma_j^{class}, j \in \{1, \ldots, k\}$, we can have a instance parameters for each sample present in the dataset, $\sigma_i^{inst}, i \in \{1, \ldots, N\}$.

This parameterization helps us to learn a curriculum over instances of a class, which is useful when instances within a class have different levels of difficulty. For instance, consider the task of classifying images of an object. In some instances, the object could be fully visible (easy), while in others, it could be occluded by other objects (hard). Another task is learning with noisy/corrupt labels. In this setting, labels of some instances would be consistent with the input (easy), while labels of some instances would not be consistent (hard). In our experiments, we show that the learning of a curriculum over instances learns to ignore the noisy samples.

We can also learn a joint curriculum over classes and instances to have the benefits of both. In this case, during training, the parameter for a data point $\mathbf{x}^i$ is set as the sum of its target's class paramter $\sigma_{y^i}$ and its own instance parameter $\sigma_i^{inst}$ i.e. $\sigma_i^* = \sigma_{y^i}^{class} + \sigma_i^{inst}$. In this setting, the gradient of the loss with respect to the logits (as in equation 3) can be expressed as $\frac{\partial L^i}{\partial z_j^i} = \frac{p_j^i - 1(j = y^i)}{\sigma_i^*}$. Since the effective parameter of an instance is formed by the addition of class and instance level parameters, the gradient for these parameters for a data point $\mathbf{x}^i$ is the same and is denoted by:

$$\frac{\partial L^i}{\partial \sigma_i^*} = \frac{(1 - p_{y^i}^i)}{(\sigma_i^*)^2} \Big( z_{y^i}^i - \sum_{j \neq y^i} q_j^i z_j^i \Big) \tag{5}$$

However note, during training, instance parameters collect their gradient from individual samples (when sampled in a mini-batch), while class parameters average the gradient from all samples of the class present in a mini-batch.

**Inference with data parameters:** As explained earlier, during training, we modify the logits of a sample with data parameters (class or instance parameters). During inference, we do not have parameters on the test set, and hence do not scale the logits with a data parameter. Not scaling the logits, has no affect on the argmax of softmax, but the classification probability is uncalibrated. If one is interested in calibrated output, calibration can be done on a held-out validation set [11]. Note this modus operandi, of not scaling logits at inference, maintains our claim: use of data parameters does not affect the model's capacity and run-time at infernece.

# 3 Experimental evaluation

In this section we first describe the implementation details of our method. Next we will show results of our method when applied for the task of image classification and detection. After that, we evaluate our dynamic curriculum framework in presence of noisy labels. Finally, we show that our framework, when applied to all random labels, acts as a strong regularizer and resists memorization. Note, since our method modifies the logits at the very end of forward pass, the gains reported below come without any additional computational overhead during training.

## 3.1 Implementation details

Optimizing data parameters $\sigma$ with gradient descent requires constraint optimization with constraint $\sigma \geq 0$. Instead, we choose to optimize in log parameterization $\log(\sigma)$, which can be mapped back using exponential mapping. Using an exponential mapping resolves log parameterization to positive domain, and allows us to perform unconstrained optimization.

In our loss function, in addition to standard $\ell_2$ regularizer on model paramters, $||\theta||^2$, we also have $\ell_2$ regularization on data parameters, $||\log(\sigma^{class})||^2$ and $||\log(\sigma^{inst})||^2$, with their contribution being controlled by weight decay parameter. This regularizer favors original softmax formulation with $\sigma = 1$, and prevents data parameters from obtaining very high values.

Unless stated otherwise, the following implementation details holds true for our experiments. For all numbers reported in this paper, we report the mean and standard deviation over 3 runs. We learn the class and instance parameters using stochastic gradient descent (SGD). Class and instance parameters are initialized with $\sigma = 1$ and optimized using gradient descent with momentum 0.9. When learning a joint curriculum over class and instances, class parameters are initialized as 1 and instance parameters are initialized as 0.001. This ensures that the sum of both parameters results in $\sigma = 1$, thereby recovering the original softmax formulation. For both sets of parameters we use separate optimizers with their respective learning rates and weight decay. The learning rate and weight decay for class parameters is set to 0.1 and $5e^{-4}$ (same as model parameters of DNN). The learning rate and weight decay for instance parameters varies depending upon the task, and is set by using the validation set. When a class or instance is not present in a mini-batch, we do not update the momentum buffer associated with the data parameter of the class or the instance respectively.

## 3.2 Learning a curriculum for image classification

In this section we demonstrate the efficacy of our method when applied to the task of image classification. We evaluate our dynamic curriculum learning framework on CIFAR100 [18] and ImageNet 2012 classification [5] dataset.

CIFAR100 dataset contains 100 classes, 50,000 images in the training set and 10,000 images in the test set. We evaluate our framework on CIFAR100 with WideResNet (depth:28, widening factor:10, dropout:0) [38]. We first reproduce the results for WideResNet[1] by setting the minibatch size, optimizer and learning rate schedule identical to the original paper [38] and report the numbers in Table 1.

ImageNet dataset contains 1000 classes, with 1.28 million training samples. We report top-1 accuracy on the validation set which consists of 50,000 images. We evaluate our framework with ResNet18 [14], we use the implementation from PyTorch's website [2]. As per the standard settings, we train the

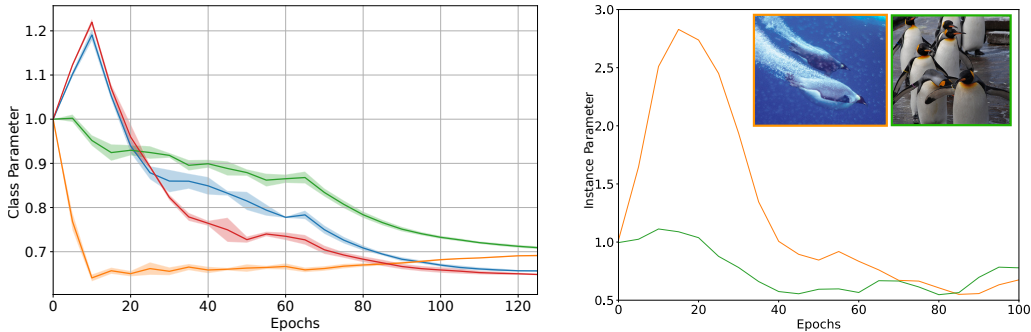

Figure 1: **Left**: Class level dynamic curriculum on CIFAR100. Curriculum learnt over classes is dynamic in nature and adapts itself for different classes. **Right**: Instance level dynamic curriculum on ImageNet. Two instances of the same class, as per their difficulty, are learnt at different points during training.

models for a total of 100 epochs, with learning rate decay of 0.1 every 30 epochs i.e. at 30, 60, and 90 epochs. Weight decay for class and instance level data parameters is set as $1e-4$ (same as model parameters) respectively. The learning rate for class and instance level data parameters is set as $0.1$ and $0.8$ respectively.

We report results for ImageNet and CIFAR100 in Table 1. As seen from the table, on CIFAR100 dataset, learning a curriculum over classes and instances lead to a statistically significant gain of $0.7\%$ over the baseline for WideResNet. On ImageNet dataset, using a dynamic curriculum translates to a gain of $0.7\%$ over the baseline. In the table, we show that learning a dynamic curriculum over classes alone performs better than baseline, but has a degradation of $0.2\%$ in accuracy when compared with class and instance level curriculum. This highlights the importance of using a curriculum over instances, and validates our hypothesis: instances within a class have varying levels of difficulty, and learning the order within a class is important. In Figure 1 (right), we plot data parameter for two instances of the same class as it evolves during training. The two instances are learnt at different points during training, as per their difficulty. For description of experiments on WebVision dataset, see section 3.4.

**Comparison with the state-of-the art:**  To the best of our knowledge, we are the first work to report gains on ImageNet dataset due to curriculum learning[3]. There are existing works which report results of curriculum learning on CIFAR100 dataset, but a direct comparison is not possible, since these works report results in different settings. Nevertheless, below, we report key results from the existing state-of-the-art:

[2] proposes a curriculum learning framework, where the sampling of data (curriculum) for SGD is based on lightweight estimate of sample uncertainty. With ResNet27 they obtain an improvement of $0.4\%$ in accuracy. Inspired from the recent work of 'Learning to Teach' [7], [36] proposes an extension, where the teacher dynamically alters the loss function for the student model. Training of a teacher model requires a separate held-out validation set and hence is not directly comparable. On CIFAR100 dataset, they obtain an improvement of $1.1\%$ over a slightly weaker ResNet-32 architecture which has $69.62\%$ baseline accuracy. [13] proposes a dynamic non-uniform sampling method for curriculum learning. They employ transfer learning to sort the training data by difficulty, and evaluate various heuristics to guide the sampling for SGD. On CIFAR100, using VGG16 without data augmentation (with a baseline accuracy of $68.1\%$), they obtain $0.6\%$ improvement in accuracy.

**Learnt curriculum is repeatable:**  To perform a qualitative evaluation, we visualize the dynamic curriculum learnt over classes in Figure 1 (left). We pick four random classes, and plot class parameters over the course of training (mean and standard-deviation over three runs). From the figure we can see that the curriculum is dynamic, and adapts to different classes. More importantly, low standard-deviation implies that the order in which classes are learnt is repeatable and intrinsic to the dataset and model. Random runs on the same dataset with different architectures lead to different curriculum.

| Dataset | Model | Baseline | DCL | Class | Instance |
|---|---|---|---|---|---|
| CIFAR100 | WRN-28-10 | $80.1 \pm 0.2$ | $\mathbf{80.8 \pm 0.1}$ | ✓ | ✓ |
| ImageNet | ResNet18 | $70.3 \pm 0.1$ | $\mathbf{71.0 \pm 0.1}$ | ✓ | ✓ |
| | | | $70.8 \pm 0.1$ | ✓ | ✗ |
| WebVision | ResNet18 | $66.3 \pm 0.1$ | $\mathbf{67.5 \pm 0.1}$ | ✓ | ✓ |
| | | | $67.1 \pm 0.1$ | ✗ | ✓ |

Table 1: Results on image classification dataset. Across different datasets and CNN architectures, using our dynamic curriculum learning (DCL) framework leads to consistent gains.

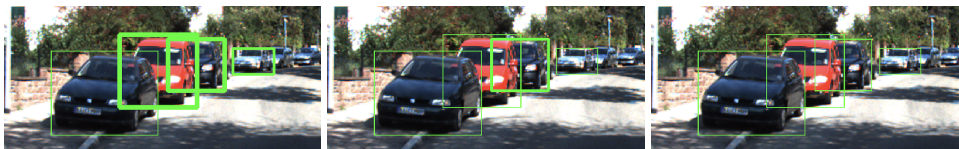

Figure 3: Dynamic curriculum over the course of training (left to right) for object detection task. Thickness of a bounding box instance is proportional to the value of parameter associated with it. The curriculum learns easier unoccluded instances first, followed by followed by partial occlusion, and in the end learns heavy occlusion.

## 3.3 Learning a curriculum for object detection

In this section we show that when applied for an object detection task, our framework is able to recover a curriculum which first learns from the unoccluded instances (easy), followed by partially occluded instances (medium) and finally learns the severely occluded instances (hard).

We apply our framework challenging KITTI dataset [9] for the task of 2D detection. The object detection benchmark from KITTI has three classes: cyclist, pedestrian and car. The training set contains 7,481 images with 2D bounding box annotations. The evaluation of 2D detectors is done in three regimes: *easy, medium and hard* defined as per the truncation and occlusion levels of objects. We use the train/ validation split provided by [3] to evaluate our performance for detecting car instances.

| Setting | Baseline | DCL |
|---|---|---|
| Easy | $92.1 \pm 0.16$ | $\mathbf{92.8}$ |
| Medium | $87.3 \pm 0.26$ | $\mathbf{87.9}$ |
| Hard | $78.0 \pm 0.6$ | $\mathbf{79.3}$ |

Figure 2: Detection mAP on KITTI.

As a baseline, we implement 2D detection using Single Shot Detector (SSDNet) [24] architecture. In SSDNet architecture, the network consists of standard convolutional layers, followed by anchors at multiple feature maps. Each anchor is assigned to either a background or to a bounding box annotation. Anchors assigned to a bounding box, predict the bounding box offset and class label. To learn a curriculum over instances, we associate a learnable parameter to each bounding box annotation. Anchors assigned to a bounding box annotation, use the value of parameter associated to that instance to rescale their logits before predicting the target class label. Therefore, if anchors assigned to a certain bounding box annotation are not able to predict the target label, the parameter associated to the bounding box instance will attain a high value. Anchors assigned to background do not have a bounding box associated to them. To mitigate this issue, for each mini-batch, we compute the mean value of instance parameter over target bounding box instances, and use that for negative anchors. This ensures that positive and negative anchors learn at the same pace, while allowing the positive anchors to learn a curriculum over different instances. We have not tuned hyperparameters on this dataset, but set learning rate for instance level parameters as 0.1, and did not use weight decay or momentum.

In Figure 2 we obtain an improvement of $0.7, 0.6$ and $1.3$ mAP in easy, medium and hard settings compared to the baseline algorithm. In Figure 3 we show how the learnt curriculum attends samples of different difficulty over the course of training.

**Comparison with the state-of-the art:** To the best of our knowledge, we are the first work to report improvements on object detection task with curriculum learning. Following works are the closest relevant state-of-the-art: [21, 33, 6] have explored the use of curriculum learning for weakly supervised object detection. To avoid getting stuck in a local minima in multiple instance learning

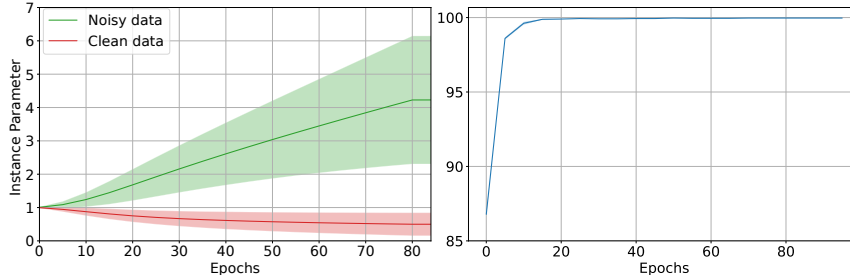

Figure 4: Visualization of dynamic instance level curriculum on noisy CIFAR100 dataset under 40% label noise. **Left**: Plot of mean and standard-deviation over instance parameters of clean and noisy data instances over the course of training. **Right**: Percentage of corrupt samples in the top 40% data points sorted by their instance parameter value. See text for details.

framework, [21] uses segmentation maps along with current bounding box proposals to define a curriculum. [33] trains an object detector using small set of training data. This detector is evaluated on large set of weakly labeled images, and is used to measure mAP per image [41]. mAP per image is used as a proxy for intrinsic difficulty of an image, and is used to define a curriculum.

### 3.4 Learning a curriculum for noisy labels

An ideal framework for learning the curriculum can be useful when some of the labels in the dataset are noisy, where the framework should prioritize learning from clean labels. In this section, we first validate our dynamic curriculum learning framework in a controlled corrupted label setting, followed by results on a real world noisy dataset.

**Results in controlled corrupted label setting** To compare with the relevant state-of-the-art, we follow the common setting in ([17, 26]) to train deep CNNs, where the label of each image is independently changed to a uniform random class with probability $p$, where $p$ is noise fraction and is set to $0.2, 0.4$ and $0.8$. The labels of validation data remain clean for evaluation. We compare our approach with two state-of-the-art approaches [17, 26] in this setting. Both of these methods assign a weight to each sample in the training set, which is used to scale the gradients of these samples during training. [17] trains an auxiliary network (MentorNet) to assign weights to data points. [26] employs meta-learning framework to learn the optimal weight of a sample.

We implement WideResNet-28-10 under settings identical to ones reported in [26]. For all of our experiments with noisy labels, the learning rate for instance parameters is set to 0.2, and accuracy is reported at 84 epochs (set by cross-validation). As seen from the results in Table 2, our method outperforms the state-of-the-art MentorNet PD[17] by $14.5\%$ on CIFAR10 and $14\%$ on CIFAR100. We also compare our results with methods (MentorNet DD [17] and robust weighting [26]) which use additional clean data to learn the curriculum. Despite the fact that our method does not use additional clean data, we outperform these methods by $2\%$ on CIFAR10 and $3\%$ on CIFAR100. In supplementary material, we perform the same analysis for $20\%$ and $80\%$ noise on CIFAR100, and show that DCL outperforms MentorNet PD[17] by $3\%$ and $22\%$ respectively.

Next, we measure the gap between our method and an oracle which learns only from the clean data. We establish the performance of the oracle by training our baseline DNN only on the clean data in each setting, i.e. in setting with $40\%$ noise, we train only on $60\%$ clean data. As it can be seen from the table, under $40\%$ noise level, the gap between our method and the oracle is only $3\%$, both for CIFAR10 and CIFAR100.

In Figure 4 (left) we plot the mean instance parameter for noisy and clean data during the course of training. As seen from the figures, over the course of training, the learnt curriculum is able to filter the clean data from noisy data, by assigning high instance parameter value to noisy instances. In Figure 4 (right) we plot the percentage of corrupt samples in top 40% of training data sorted by their instance parameter value. As seen from the plot, within 20 epochs, 95% of the noisy instances attain instance parameter values greater than all clean samples. For results under $20\%$ and $80\%$ noise level, see supplementary material.

| | Additional Clean Data | CIFAR-10 | CIFAR-100 |
|---|---|---|---|
| MentorNet DD [17] | Yes | 88.7 | 67.5 |
| Robust Weighting [26] | Yes | $86.92 \pm 0.19$ | $61.34 \pm 2.06$ |
| Baseline [26] | No | $67.97 \pm 0.62$ | $50.66 \pm 0.24$ |
| Reed Hard [26] | No | $69.66 \pm 1.21$ | $51.34 \pm 0.17$ |
| S Model [26] | No | $70.64 \pm 3.09$ | $49.10 \pm 0.58$ |
| MentorNet PD [17] | No | 76.6 | 56.9 |
| DCL (ours) | No | $\mathbf{91.10 \pm 0.70}$ | $\mathbf{70.93 \pm 0.15}$ |
| Baseline on clean data (oracle) | No | $94.24 \pm 0.15$ | $74.18 \pm 0.19$ |

Table 2: Performance of our method under uniform 40% label noise on train set. Dynamic curriculum learning (DCL) outperforms the state-of-the-art methods including methods (top 2 rows) which use additional clean data. Bottom row indicates performance of baseline DNN trained on clean labels.

**Results on noisy dataset from web**   In this section we will show results on the challenging WebVision 2017 dataset [22], a large scale dataset, which has corrupted labels and is extremely imbalanced. WebVision 2017 dataset is constructed by crawling Google image search and Flickr using 1000 classes from ImageNet as queries. It contains 1000 classes, with 2.4 million training images, without any human annotation. The dataset provides 50,000 manually-labeled images for evaluation.

We conducted experiments using ResNet18 with the same hyper-parameters as we have used for ImageNet experiments in the paper and report results in Table 1. Our baseline, the standard DNN training of ResNet18 obtained $66.3 \pm 0.1\%$ as top-1 accuracy. Since the noise present in dataset is at instance level, first we evaluate the use of instance level curriculum. Using an instance curriculum leads to an improvement of $0.8\%$. Next, we evaluate the use of joint class and instance level curriculum. Interestingly, even though noise present in this dataset is at instance level, learning a joint curriculum improves over instance level curriculum, and leads to an overall gain of $1.2\%$ over the baseline.

### 3.5   Curriculum learning with all random labels

Recent work [39] has shown that when presented with a dataset containing all random labels, standard DNNs are able to memorize the entire training dataset. They evaluated the standard regularization methods such as weight-decay, dropout, data-augmentation, and found them to be ineffective to prevent memorization. We replicate their experimental setup and findings using VGG16 [28] (baseline) on CIFAR100 dataset. In 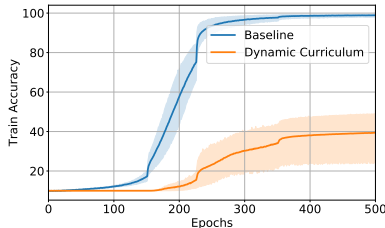 the figure on the right, we plot the training accuracy curves for baseline and baseline with our dynamic curriculum. As seen from the plot, using our dynamic curriculum learning formulation resists memorizing the corrupt training data. As explained in Section 2.1, when data-points of a certain class are misclassified, the gradient update will increase the corresponding class parameter. In this setting, where labels of images are random, over the course of training, the class parameter for all the classes keeps increasing, effectively decaying the magnitude of gradient update on the training set (see equation 4).

## 4   Related work

Curriculum learning has been an active topic of research in the machine learning community and has been used in various problems [4, 10, 15, 21, 27, 31, 32]. In this section, we give a brief overview of related work most relevant to the material we present in the paper. For a brief overview of curriculum learning and a theoretical treatment, we refer the reader to [34].

In the early works of curriculum learning [1, 29], the curriculum was pre-determined and fixed during the course of optimization. To address this limitation, [19] proposed Self Paced Learning (SPL) framework, where the curriculum is optimized jointly with the model parameters. In SPL [19], the data points are assigned a weight variable, which are updated along with the model parameters using alternate minimization. More specifically at each iteration, weights of samples with a loss higher than a pre-defined threshold $\lambda$ are set to 0. Over the course of training, while gradually increasing $\lambda$, more samples are included in training from easy to hard in a self-paced manner. SPL has been widely adopted and applied to various problems [20, 23, 25, 30]. Similar to SPL, our method learns a

dynamic curriculum and mitigates the issue of a using a pre-determined curriculum. Earlier works in curriculum learning and SPL perform discrete sampling of data, which could lead to local minima. In comparison, our method performs soft-differentiable sampling of data. Recently, several works have been proposed to obtain better weighting strategies [7, 16, 17, 40] for SPL framework. Learning weight for each sample amounts to learning the scale for the gradient update of each sample. In contrast, in our method, learning the data parameters amounts to learning the loss function specific to each data point and class. Another major difference between our method and SPL is that, the majority of SPL methods use the loss of a data-point as a proxy for establishing its hardness with respect to the current model. This heuristic when applied to deep neural networks (DNNs) might be problematic, since DNNs can easily memorize hard examples (e.g random labels[39]), making the loss of a sample decorrelated with the intrinsic hardness of the sample.

Recent works have explored meta-learning for modifying the loss function dynamically [36], to re-weight instances to enable learning with noisy labels [17, 26, 36] and to accelerate training of DNNs [7]. These methods involve training a teacher on a task, and then using the teacher to train the student on the target task. In contrast, in our method, the parameters for instances and classes (viewed as teacher) and the parameters of the model (viewed as student) are trained jointly. Doing so ensures that the learnt curriculum is consistent with the current state of the model, and does not require a held out dataset.

Curriculum learning has also been explored in the context of learning with noisy labels [12, 17, 37]. MentorNet[17] trains an additional network for weighing samples in a noisy train set. Guo *et al.* [12] propose a novel curriculum learning framework by measuring the data complexity using clustering density. They apply their method on large-scale weakly-supervised web images and obtain state-of-the-art results. For a comprehensive overview on label noise and noise robust algorithms we refer the reader to [8].

## 5 Conclusion

In this work, we have introduced a new family of parameters termed "data parameters". We have shown that data parameters can be learnt using gradient descent, and doing so amounts to learning a dynamic curriculum. Specifically, we equip each class and training data point with a learnable parameter (*data parameters*), which governs their importance during different stages of training. Along with the model parameters, the data parameters are also learnt with gradient descent, thereby yielding a curriculum which evolves during the course of training. More importantly, post training, during inference, data parameters are not used, and hence do not alter the model's complexity or run-time at inference. We apply this dynamic curriculum learning framework to image classification and object detection tasks, and show that our approach leads to consistent gains over the baseline DNNs. When applied to a noisy dataset, the dynamic curriculum priortizes learning from clean data, while effectively ignoring noisy data. Finally, when presented with dataset containing random labels, our framework resists memorizing the training data unlike standard DNNs.

## Footnotes

[1]Authors report results as median run of 5 runs. We reimplement their method and report mean and standard deviation over three runs.

[2]https://github.com/pytorch/examples/tree/master/imagenet

[3][17] reports numbers on ImageNet, but they train on a dataset twice the size of ImageNet containing noisy labels. In section 3.4, we make an explicit comparison with [17] for the task of learning with noisy labels.

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
