[Supplementary Material]

# Data Parameters: A New Family of Parameters for Learning a Differentiable Curriculum Supplementary Material

**Shreyas Saxena**
Apple
shreyas_saxena@apple.com

**Oncel Tuzel**
Apple
otuzel@apple.com

**Dennis DeCoste**
Apple
ddecoste@apple.com

## Results for CIFAR100 under different noise ratio

| Model | Additional Clean Data | Noise = 20% | Noise = 80% |
|---|---|---|---|
| MentorNet DD [1] | Yes | 73.0 | 35.0 |
| Baseline | No | 60.0 | 8.00 |
| Forgetting [1] | No | 61.0 | 16.0 |
| Self-paced [1] | No | 70.0 | 13.0 |
| Reed Soft [1] | No | 62.0 | 8.0 |
| MentorNet PD [1] | No | 72.0 | 14.0 |
| DCL (ours) | No | $\mathbf{75.68 \pm 0.12}$ | $\mathbf{35.8 \pm 1.0}$ |
| Baseline on clean data (oracle) | No | $77.79 \pm 0.42$ | $58.32 \pm 0.53$ |

Table 1: Performance of our method on CIFAR100 under under 20% and 80% uniform label noise with WideResNet28-10 model. Test accuracy shown in percentage.

## References

[1] Lu Jiang, Zhengyuan Zhou, Thomas Leung, Li-Jia Li, and Li Fei-Fei. Mentornet: Learning data-driven curriculum for very deep neural networks on corrupted labels. In *ICML*, 2018.