[Reviews · NeurIPS 2019]

Reviewer 1



Originality: The idea is interesting, and to the best of my knowledge hasn't been tested before. However, applying gradient descent to update parameters is not very original. Clarity: Overall, the paper is well written. Te structure can be improved however. For example, having the related works sections right before the conclusion is quite disturbing. Quality: The description of the method is sound and technically correct. The experimental section is well designed and fairly assess the method. I particularly appreciate the experiments with noisy labels. However, I have some concerns. Can the authors comment on the importance of the instance weights? Adding a trainable weight per sample can substantially increase the size and capacity of the model. This can make the comparison to the baselines unfair. Moreover, a missing analysis in the paper is to track and try to analyze the parameters in order to check if they actually do what they are expected to, i.e. prioritize the easy samples in the beginning of training and progressively evolve towards uniform importance. Significance: The experimental results are extensive and show significant improvement over the state-of-the-art models in several settings and for several datasets. This proves the the interest of the method, but again, it is not clear to me how much of this performance gain is due to the curriculum learning mechanism.

Reviewer 2



Originality: The intuition of the model is similar to existing curriculum base model (e.g. MentorNet), but the approach to estimate the weight of each example is new. In this paper, the authors proposed to learn the weight of each instance/class with a gradient descent. The related work section is clear and explain the difference with existing approaches. Quality: The proposed dynamic curriculum learning framework is validated on image classification and object detection tasks. In particular, the proposed curriculum based approach outperforms existing curriculum based approaches like MentorNet. The model is also accurate when it learns with noisy labels because it learns to ignore noisy training examples. Clarity: The submission is clearly written and well organised. Significance: A lot of approaches have been proposed recently to speed-up the training of deep models. I think that using curriculum based models is an interesting direction to speed-up the convergence. I also think that designing models that are robust to noisy labels is an important problem because a lot of real data contains noisy labels. --------------------------------------------- The rebuttal addresses my concerns and I think it is a good paper.

Reviewer 3



The paper presents a novel approach to curriculum learning, by introducing two new sets of parameters to learn, one per class and one per example, which correspond to temperatures in the softmax classification layer, and can easily be trained by gradient descent. When considering the class-based version (equation 1), I wonder why the model without that extra set of parameters cannot learn the same thing through the existing weights (it's just a scaling of the logit, after all). For instance-level temperatures, it makes more sense but again, this can only work with small enough datasets where each example is expected to be seen many times; in the context of a very large dataset, it's unlikely that each example would be seen enough by the model to learn a relevant temperature. Having a temperature parameter that is a function of the example (and maybe the label) could alleviate this (but this might become similar to other competing approaches, no?) I would suggest numbers given in the text page 5 around line 170 be put in the relavant Table 1 for ease of comparison. I didn't understand the difference suggested on page 8, line 285-286.

[Author Response · NeurIPS 2019]

We thank the reviewers for their valuable comments. We are glad that reviewers noted our paper as novel (R1: "idea is interesting .. and hasn't been tested before", R3: "approach to estimate weight of example is new", R4: "novel approach to curriculum learning by introducing new sets of parameters"), and have appreciated our results (R1: "results are extensive, and show significant improvement in several datasets", R3:"outperforms existing curriculum learning based approaches"). Below, we provide clarifications to the points they have raised, and provide additional experiments requested by the reviewers for improvement of rating.

**Reviewer 1:**

– **Requested Improvement** *"Decouple the effect of capacity increase and curriculum learning"*: We would like to clarify that the temperature parameters for class and instances are not parameters of the model. They are used only during training to modify the loss function. The architecture used for inference in our model and the baseline are identical, therefore the capacity (number of model parameters) is exactly the same. Hence, the gains we obtain on different datasets and tasks are due to curriculum learning. Thanks for pointing out a potential source of confusion; we will clarify this point in revision. We will also move related works section as suggested.

– *"Applying gradient descent to update parameters is not very original"*: Introducing trainable temperature parameters for instances and class in a dataset, and optimizing them through gradient descent is our original contribution.

– *"Comment on the importance of instance level parameters"*: In Table 1 (in paper) we present an ablation study where instance level curriculum provides additional improvement over class level curriculum on ImageNet and CIFAR100. In addition, the improvements on noisy datasets are solely due to instance level curriculum, since the per sample noise can only be mitigated by instance level curriculum. The reason class level curriculum can not help in this case, is because it assumes homogeneous difficulty across samples within a class.

– *"Missing analysis in paper is to track and analyze parameters"*: Please see Figure 3, and Figure 4 (left) in paper, where we have tracked and analyzed temperature parameters. Figure 3: For learning a detector, curriculum learns easier unoccluded instances first, followed by partial occlusion, and finally heavy occlusion. Figure 4 (left): Shows that the temperature of noisy samples keeps increasing over the course of training, hence decaying their contribution to learning process. We agree that this issue is important in the field of curriculum learning. For final version, we will provide more explicit examples demonstrating the learnt curriculum.

**Reviewer 3:**

– **Requested Improvement 1** *"It could be interesting to show results on the large WebVision Benchmark"*:

|  | R18 | R18 + DCL |
|---|---|---|
| Top-1 Acc | 66.3 | **67.6** |

As you suggested, we conducted experiments using ResNet18 with the same hyper-parameters as we have used for ImageNet in the paper. As shown in table (left), **we obtain an absolute improvement of $1.3\%$ in top-1 accuracy on this challenging dataset** which in addition to being a large-dataset, has noisy labels, and is extremely imbalanced.

– **Requested Improvement 2** *"Would proposed curriculum change robustness to adversarial attacks"*:

| Metric | R18 | R18 + DCL |
|---|---|---|
| Top-1 Acc Adv. | 44.3 | **46.0** |

Thanks for pointing us in this direction. As you suggested, we conducted an initial investigation with untargeted FGSM attack (Goodfellow et al., 2014) on ImageNet and found this direction to be promising. As shown in table (left), **model trained with curriculum obtains $1.7\%$ higher accuracy (post adversarial attack)** compared to baseline.

– *"Curriculum based methods is an interesting direction to speed-up convergence"* : While speeding up training of DNNs was not our explicit goal, we did, thanks to your comment, an analysis for experiments reported in paper on ImageNet. We measured the relative reduction in number of epochs for our method to achieve the same accuracy as the baseline at various points during the training. **On average, our method requires $20\%$ fewer epochs.**

**Reviewer 4:**

– **Requested Improvement** *"Results on larger training sets or datasets with large number of classes"*: In addition to ImageNet, we conducted new experiments on WebVision dataset (2.3 million training images) and obtain significant gains. Please see the first table above. When we analyzed temperature trajectories over the course of training (eg. Figure 1 right in paper), within the first few epochs, temperature of the hard instance (orange curve) peaks, decaying its contribution to learning. Empirically, most of the temperature variation for instances occurs early on during optimization (<30 epochs). Visiting the same data point 30 times (in multiple datasets of the scale of millions of data-points) was sufficient to learn the instance level temperature parameters. Nevertheless, we agree for datasets which contains of billions of training samples, and training loop might visit a data point only once or twice, alternative formulations should be explored.

– *"Why model without temperature parameters for class can not learn the same loss function?"*: Thank you for pointing this out, we can see this as an easy source of confusion. Unlike scaling each logit with temperature of its respective class (which could indeed be absorbed in weights), in our formulation, we scale all the logits of a sample, with temperature of the target class. In other words, in paper's Eq 1, notice that subscript of class temperature parameters is $y_i$ (target label of sample $i$) in the denominator ($\sum_j \exp(z_j^i / \sigma_{y_i}^{class})$) and not $j$. This cannot be absorbed by scaling the weights of the model. We will also clarify the difference suggested on page 8.

[Meta-Review · NeurIPS 2019]

This work proposes an optimization scheme for learning a curriculum over classes or training samples. The importance of each sample/class is reflected by a learnable parameter that is learned by gradient descent simultaneously with network weights. The proposed scheme particularly shows its advantage in noisy data as demonstrated empirically. All reviewers find their concerns well-addressed in authors' response, and they all find the paper a solid and interesting work.